# Neurodevelopmental Changes in the Guinea Pig Brain Caused by Time-Limited Complete Vitamin C Deprivation

**DOI:** 10.3390/nu17213484

**Published:** 2025-11-06

**Authors:** Ivan Čapo, Ilija Andrijević, Nataša Čapo, Milan Popović, Ivan Milenković, Radomir Ratajac, Dejan Vranješ, Dragana Milutinović, Dragana Simin, Slobodan Sekulić

**Affiliations:** 1Department of Histology and Embryology, Faculty of Medicine Novi Sad, University of Novi Sad, 21000 Novi Sad, Serbia; ivan.capo@mf.uns.ac.rs (I.Č.); milan.popovic@mf.uns.ac.rs (M.P.); 2Centre for Pathology and Histology, University Clinical Centre of Vojvodina, 21000 Novi Sad, Serbia; 3Department of Nursing, Faculty of Medicine Novi Sad, University of Novi Sad, 21000 Novi Sad, Serbia; dragana.milutinovic@mf.uns.ac.rs (D.M.); dragana.simin@mf.uns.ac.rs (D.S.); 4Institute for Pulmonary Diseases of Vojvodina, 21204 Sremska Kamenica, Serbia; 5Department for Epidemiology, Faculty of Medicine, Novi Sad, University of Novi Sad, 21000 Novi Sad, Serbia; natasa.capo@mf.uns.ac.rs; 6Department of Prevention of Rabies and Other Infectious Diseases, Pasteur Institute Novi Sad, 21000 Novi Sad, Serbia; 7Oncology Institute of Vojvodina, 21204 Sremska Kamenica, Serbia; 8Department of Neurology, Medical University of Vienna, 1090 Vienna, Austria; ivan.milenkovic@meduniwien.ac.at; 9Department for Food Safety and Drug Analysis, Scientific Veterinary Institute “Novi Sad”, 21000 Novi Sad, Serbia; ratajac@niv.ns.ac.rs; 10Faculty of Medicine Novi Sad, University of Novi Sad, 21000 Novi Sad, Serbia; dejanvranjes@uns.ac.rs; 11Department of Neurology, Faculty of Medicine Novi Sad, University of Novi Sad, 21000 Novi Sad, Serbia; slobodan.sekulic@mf.uns.ac.rs; 12Institute for Child and Youth Health Care of Vojvodina, 21000 Novi Sad, Serbia

**Keywords:** ascorbic acid, brain, deficiency diseases, neuropathology, guinea pig

## Abstract

Background/Objectives: The guinea pig is a unique experimental model because of the evolutionary loss of the GULO gene, which encodes an enzyme involved in vitamin C synthesis. Since vitamin C plays an essential role in collagen biochemistry, numerous studies have investigated the effects of pre- and postnatal vitamin C deficiency. However, only a few studies, including ours, have indicated a possible link between vitamin C deprivation and potential weakening of the basement membrane, which may lead to significant alterations in brain structure. Methods: The experiment included guinea pig foetuses completely deprived from the 10th (E2 group) and the 20th (E1 group) to the 50th day of intrauterine life. Tissue samples from the cerebrum and cerebellum were taken for biochemical, molecular, and immunohistochemical analyses. Results: In the E2 group alone, we found marked gross changes: cerebral bleeding, porencephaly, and a lissencephalic cerebellar surface. Microscopic examination revealed diffuse bleeding in the cerebrum along with a loss of neurons in the area of the defect, specifically in the E2 group. The complete maturation of ectopic neurons characterised dysplastic changes in the cerebellum. Hydroxyproline analysis of both the cerebrum and cerebellum showed no significant differences among the E1, E2, and control groups. However, decreased expression of COL1, COL4A1, and SLC23A1 was observed solely in the cerebellar tissue of the E1 group. Conclusions: The morphological, biochemical, and molecular results represent preliminary associations with vitamin C deficiency, but require further validation.

## 1. Introduction

In addition to humans, several mammals, including higher primates, guinea pigs, capybaras, bats, certain birds, and various species of fish, are unable to synthesise vitamin C endogenously [1,2]. It is believed that a mutation occurred during evolution that led to the loss of the GULO gene, which encodes the enzyme L-gulono-γ-lactone oxidase [3]. The lack of this enzyme directly affects the synthesis of vitamin C, which, in turn, is involved in the hydroxylation of proline during collagen synthesis [4]. This clearly defined pathomechanism represents a long-forgotten clinical entity called scurvy. Compromised collagen biosynthesis leads to systemic connective tissue breakdown, with classic symptoms such as bleeding gums, easy bruising, poor wound healing, and musculoskeletal pain [5]. Although forgotten, scurvy is again on the rise in developed countries, mainly due to poor diets high in processed foods and low in fresh vegetables and fruits like red peppers (125–150 mg in medium size), Broccoli (39,2 mg in 1/2 cup) or oranges (63.5 mg in medium size), strawberries (97 mg in 1 cup) and kiwifruit (70.5 mg in medium size). At-risk groups include alcoholics, smokers, individuals with eating disorders like anorexia, the elderly, and patients with chronic gastrointestinal diseases such as Crohn’s disease. Smoking notably reduces vitamin C absorption and increases the body’s requirement for vitamin C [6,7].

The most frequently cited clinical symptom of vitamin C deficiency is bleeding from small blood vessels. It occurs due to increased fragility caused by aberrant collagen IV production, a fundamental structural element of the basement membrane [8,9]. In the brain, the pial basement membrane (PBM) serves as an essential scaffold for the development of the cerebral cortex. Composed of the pia mater and the glia limitans, it provides a pathway for migrating neuroblasts and organises radial glia fibres. During brain development, the preservation and integrity of the PBM are key factors for proper neuronal migration [10,11]. Namely, the loss of appropriate attachment of the radial glial to PBM results in a disturbance of the glial scaffold [12]. As a possible cause of the PBM rupture, researchers report disturbance of the synthesis of its structural constituents, such as dystroglycan, collagen IV, laminin, and others [13,14,15,16]. A notable example is the gene mutation responsible for collagen type IV synthesis, which leads to ocular dysgenesis, neural migratory defects, and congenital muscular dystrophy [17].

Guinea pigs are ideal experimental animals for studying vitamin C deficiency because they lack the GULO gene due to evolutionary changes [1,2] (p. 1). Several studies have highlighted the significance of prenatal deprivation and its consequences, such as unsuccessful matings, a higher incidence of foetal reabsorption or growth restriction and premature birth [18,19,20]. Changes observed in the brain tissue primarily included a reduction in the number of neurons [21,22]. However, the most significant focus was on the biochemical alterations resulting from the lack of antioxidant protection provided by Vitamin C [23,24]. However, all the studies mentioned above included only partial vitamin C deprivation, not complete deprivation.

To highlight the importance of PBM structural integrity during brain development, we developed a novel animal model to examine PBM damage [25]. We hypothesised that foetal guinea pigs lacking exogenous vitamin C could reflect weakness and ultimately rupture the PBM. Our study specifically examined the effects of complete vitamin C deprivation during a time-limited period of prenatal life in guinea pigs. The duration and timing of the deprivation are possibly associated with minor to extensive changes, both macroscopic and microscopic, particularly in the brain tissue, characterised by marked neuronal dysplasia [25].

Based on a previous study [25], here, we present an extensive immunohistochemical analysis of cerebrum and cerebellum, focusing on neural and glial differentiation in intensely dysplastic areas. We also conducted biochemical and molecular studies in the animal model presented here.

## 2. Materials and Methods

### 2.1. Animals and Environment

We used albino guinea pigs (*Cavia porcellus*) aged 4 to 6 months, obtained from the Faculty of Medicine in Novi Sad. Animals were housed in plastic containers measuring 400 mm (width) × 1000 mm (length) × 300 mm (height), arranged in a harem system with 3 or 4 females per male. Artificial cycles provided 12 h of light and 12 h of dark. The ambient temperature was maintained at 23 degrees Celsius, and the air was circulated 6 to 10 times per hour. The presence of sperm in smears from the vaginal introitus indicated the first day of gestation, after which the dams were housed separately from the males. In guinea pigs, gestation lasts between 62 and 68 days. The experimental study involved 30 female animals that maintained their pregnancies until they were euthanised. The animals were divided into three groups: a control group, Control (n = 10), and two experimental groups, E1 (n = 10) and E2 (n = 10). Experimental animals that could not achieve or maintain pregnancy during the experiment were excluded and replaced with randomly selected new dams. A clear sign that an abortion had occurred in the dam (before E50) was the appearance of blood in the vaginal opening and sudden weight loss. Final flow of dams was: control group, enrolled (n = 0), excluded with reasons (n = 0), analysed (n = 10); E1 group, enrolled (n = 11), excluded with reasons (n = 1), analysed (n = 10); and in E2 group, enrolled (n = 19), excluded with reasons (n = 9), analysed (n = 10).

### 2.2. Experimental Protocol

All animals were maintained on a standard commercial pellet diet free from vitamin C. Control animals were given unlimited access to water containing vitamin C (50 mg/L). Until vitamin C restriction, females in the experimental groups had the same diet as those in the control group. Vitamin C deprivation began on the 10th day for group E2 and on the 20th day for group E1, continuing until the 50th day of pregnancy (E50) when all animals were euthanised. Each group included 10 litters, each with 3 or 4 foetuses. The final number of foetuses was as follows: The Control group had 32, E1 had 33, and E2 had 31. From each group, six litters were randomly selected: 1 foetus per litter was taken for biochemical and molecular analysis, and 1 or 2 foetuses per litter for histological analysis. All foetuses from the remaining four litters in each group underwent histological analysis. Appendix A (Appendix A) details the distribution.

### 2.3. Fixation and Histology Procedure

On the E50, foetuses were removed by caesarean section under urethane anaesthesia and transcardially perfused with Zamboni’s fixative. After fixation, the central nervous structure (cerebellum, cerebrum) was removed from the skull and immersed in a fixative for 24 h at 4 °C. For this study, we present data derived from the cerebellum and cerebrum. For histological analysis, we cut a 5 mm-thick frontal section of cerebrum from −4.08 to −5.28 mm from Bregma and a midsagittal section from the region of the cerebellar vermis. After appropriate dehydration, the section was embedded in paraffin (Histowax, Gothenburg, Sweden) and cut on a rotary microtome (Leica, Wetzlar, Germany). Slides were stained with hematoxylin and eosin (H&E) and Perls Prussian blue for the detection of iron presence in the tissue. Methods for immunohistochemical staining included primary antibodies (Table 1) using DAKO EnVision© detection kit, peroxidase/DAB, rabbit/mouse (Dako, Santa Clara, CA, USA) for anti-NeuN, anti-Olig2, anti-GFAP and anti-TPPP; EXPOSE Mouse and Rabbit Specific HRP/DAB Detection IHC kit (Abcam, Cambridge, UK) for anti-DCX, anti-nestin, anti-S100, anti-synaptophysin, and anti-calbindin D-28k. To detect anti-MBP, we use a Donkey polyclonal Secondary Antibody to Rat IgG-H&L (HRP) (Abcam, Cambridge, UK) and DAB as the chromogen. Before applying all antibodies except anti-calbindin and anti-S100, we performed antigen retrieval using citrate buffer (pH 6.0) in a microwave oven at 850 W for 20 min. Mayer’s hematoxylin was used as a counterstain for immunohistochemistry, followed by mounting and coverslipping (Bio-Optica, Milan, Italy) for slides. Histology slides were converted to digital slides using a high-resolution digital slide scanner, the NanoZoomer 2.0-HT (C9600-13, Hamamatsu Photonics K.K., Shimokanzo, Japan). Each digitalised slide was visually inspected using the NDP.view software (NanoZoomer Digital Pathology Image, Hamamatsu Photonics K.K., Shimokanzo, Japan).

### 2.4. Determination of Hydroxyproline Concentration

The concentration of hydroxyproline in tissue samples was measured using the modified Hoffman method. Tissue samples were frozen at −80 °C until analysis, then thawed, weighed, and homogenised in distilled water (100 μL of dH_2_O per 10 mg of tissue). The homogenate was mixed with concentrated hydrochloric acid (100 μL) and hydrolysed at 120 °C for 3 h. After hydrolysis, the mixture was centrifuged for 3 min at 10,000× *g*, and 10 μL of the supernatant was placed in a 96-well microtiter plate. The samples were heated to 60 °C until evaporation, then the buffered Chloramine T reagent (100 μL) was added for oxidation, and the mixture was incubated at room temperature for 5 min. Perchloric acid (50 μL) was added, and the mixture was incubated for an additional 5 min. Finally, 50 μL of 20% DMAB reagent was added, and the mixture was incubated at 60 °C for 90 min. Absorbance was measured at 560 nm using a microtiter plate reader (MultiSkan FC, Thermo Scientific, Waltham, MA, USA). Each tissue sample was analysed in triplicate, and the results were expressed as μg hydroxyproline per milligram of analysed tissue.

### 2.5. Isolation of RNA

RNA was isolated from adequately sampled fresh cerebellar tissues of all three experimental groups using TRIzol Reagent (Invitrogen, Waltham, MA, USA). Tissues were homogenised and lysed, then 0.2 mL of chloroform was added to separate the phases. After shaking for 15 s and incubating for 2–3 min at 25 °C, samples were centrifuged at 12,000× *g* for 5 min at +4 °C. The supernatant containing RNA was transferred to new DNase- and RNase-free tubes, and the RNA was precipitated with 0.5 mL of isopropyl alcohol for 10 min at 25 °C, followed by another centrifugation. The RNA precipitate was washed with 1 mL of 75% ethanol and centrifuged at 7500× *g* for 10 min at 4 °C. The residual ethanol was evaporated at 30 °C, and the RNA was dissolved in 40–50 μL DEPC-treated H_2_O. Residual genomic DNA was removed using the Turbo DNA-free™ Kit (Ambion, Austin, TX, USA). RNA concentration was measured with a Qubit 3 fluorimeter and the Qubit RNA Broad Range Assay Kit (Thermo Fisher Scientific, Waltham, MA, USA).

### 2.6. Reverse Transcription Reaction

Isolated RNA cannot be used for PCR since Taq polymerase requires single-stranded DNA as a template. Hence, mRNA is converted into complementary DNA (cDNA) using a High-Capacity cDNA Reverse Transcription Kit (Applied Biosystems, Waltham, MA, USA) and an RNase inhibitor. The RNA input must be between 100 ng and 2 μg, with a total reaction volume of 20 μL, consisting of 10 μL of sample and 10 μL of master mix containing RT buffer, dNTP Mix, RT primers, reverse transcriptase, RNase inhibitor, and nuclease-free water.

### 2.7. Relative Quantification of mRNA in Real-Time PCR

Real-time PCR enables monitoring of amplification after each cycle using a Real-Time PCR 7500 FAST System (Applied Biosystems, Waltham, USA). The conditions include 50 °C for 2 min, 95 °C for 10 min, followed by 40 cycles at 95 °C for 15 s and 60 °C for 1 min. All primers, including those for Cavia porcellus glyceraldehyde-3-phosphate dehydrogenase (GAPDH), were pre-designed (see Table 2).

SYBR Green was used as the fluorescent dye, alongside 2.5 μL of diluted cDNA and Power SYBR Green PCR Master Mix (Applied Biosystems, Waltham, MA, USA). Each experimental sample was prepared in duplicate. Results were analysed using SDS software version 1.4 (Real-Time PCR 7500 FAST, Applied Biosystems, Waltham, MA, USA).

### 2.8. Statistical Analysis

Statistical analysis was performed using IBM SPSS statistical software, version 26.0 (IBM Corp., Armonk, NY, USA). Data were reported as the mean ± standard deviation (SD) or standard error of the mean (SEM). The normality of continuous variable distributions was assessed using the Kolmogorov–Smirnov and Shapiro–Wilk tests. In addition, the choice of alternative methods (nonparametric techniques) was based on variable type, coefficient of variance, and the results of the Levene test for homogeneity of variances. Because the assumptions for the chi-square test were not fully met, Fisher’s exact test (Monte Carlo method, two-sided) was applied to obtain precise significance values for differences in the frequency of intracranial haemorrhages among the experimental groups. A one-way analysis of variance (ANOVA) was used to compare groups for hydroxyproline concentrations. Post hoc testing for ANOVA was performed using Tukey’s test. For the analysis of mRNA relative concentration, an independent-samples T-test or its nonparametric alternative, the Mann–Whitney U test, was used. For each test, the corresponding effect size was calculated (Cohen’s d for *t*-tests, r for non-parametric tests). Where applicable, 95% confidence intervals (CIs) for the effect sizes were also reported to indicate the precision of the estimates. The difference between groups was considered statistically significant for a *p*-value less than 0.05 (*p* < 0.05).

## 3. Results

### 3.1. Morphological Analysis of Cerebellar and Cerebral Tissue

#### 3.1.1. Macroscopic Cerebellar and Cerebral Analysis

As we presented in our previous publication [25], in both the control and E1 groups, we observed normal development and anatomical arrangement of the cerebellar folia. In all foetuses from the E2 group, the cerebellum showed a minor to severe anatomical folia disarrangement with brain surface flattening and loss of the border between hemispheres and vermis (Figure 1).

A gross analysis of the cerebrum in guinea pigs at the E50 age revealed no significant differences between the control and E1 groups. The cerebral hemispheres were appropriately sized and symmetrical. However, in a subset of individuals from the E2 group (n = 11), there were observable areas of extensive confluent subarachnoid haemorrhages (SAH), accompanied by focal petechial haemorrhages. In some of these bleeding areas, the cerebral cortex was absent, resulting in deeper and shallower defects, which were noted as prosencephaly (see Figure 1).

#### 3.1.2. Microscopic Cerebral Analysis

At E50 in the Control group, the cerebral cortex was well developed, with clear laminar neuronal organisation (Figure 2). Almost all cortical neurons were mature, postmitotic, and exhibited high NeuN expression. At the same time, in the neuroepithelial layer, we still identified a small number of cells (Figure 2B, arrows). Selective calbindin positivity was mainly detected in small and large pyramidal neurons, lamina III and V, respectively (Figure 2). A high degree of neuronal differentiation was also confirmed with diffuse synaptophysin positivity in cerebral grey matter (Figure 2D). The glial marker GFAP was primarily expressed in the neuroepithelial layer, whereas S100 showed focal positivity in glial cells across all laminas (Figure 2E,F). Although Olig2 single cell positivity was identified diffusely in all laminas (Figure 2G), a white mass showed an intense staining on myelin basic protein (MBP) (Figure 2H). Elements of intraparenchymal and subarachnoid bleeding have not been identified in any of the foetuses from the Control group (Table 3).

Statistically significant differences in the occurrence of haemorrhages among the groups were observed in the cerebral cortex (χ^2^(2) = 40.995, *p* < 0.001; Fisher’s exact test, Monte Carlo method), hippocampus (χ^2^(2) = 42.367, *p* < 0.001), thalamus and hypothalamus (χ^2^(2) = 40.995, *p* < 0.001), cerebellum (χ^2^(2) = 52.046, *p* < 0.001), and in the case of extra-axial (subarachnoid) haemorrhage (χ^2^(2) = 29.066, *p* < 0.001).

Histological evaluation of brain tissue in both experimental groups indicates abnormalities in the basal membranes of blood vessels and the pial membrane. Focal, intraparenchymal bleeding in the cerebral cortex, hippocampus, thalamus, hypothalamus and cerebellum (Table 2) was observed in a small number of foetuses from the E1 group (up to 30%). In most cases, it was a focal fresh bleeding with visible erythrocytes around the small capillaries in the surrounding nervous tissue (Figure 2S). However, hemosiderophages, as a clear indicator of old bleeding (Figure 2T), were also found. For the E1 difference, intraparenchymal haemorrhage was observed in all of the above-mentioned brain structures in all E2 groups (Table 3). Statistically significant differences in the occurrence of haemorrhages among the groups were observed in the cerebral cortex (χ^2^(2) = 40.995, *p* < 0.001; Fisher’s exact test, Monte Carlo method), hippocampus (χ^2^(2) = 42.367, *p* < 0.001), thalamus and hypothalamus (χ^2^(2) = 40.995, *p* < 0.001), cerebellum (χ^2^(2) = 52.046, *p* < 0.001), and in the case of extra-axial (subarachnoid) haemorrhage (χ^2^(2) = 29.066, *p* < 0.001). In over 50% of E2 foetuses, we identified an extensive, confluent subarachnoid bleeding which deeply penetrated and destroyed underneath the cerebral wall without clear communication with the ventricle. In some cases, the lesion even reached the VI lamina of the cerebral cortex. Histologically, these porencephalic lesions resemble a cystic cavity filled with fresh blood and limited by the surrounding brain tissue, with an encephalomalatic wall composed of degenerating neurons, numerous macrophages (foamy histiocytes), and increased numbers of glial cells (Figure 2Q,R). The bordering neuronal tissue was characterised by a markedly reduced density of NeuN-positive neurons (Figure 2J), decreased synaptophysin immunohistochemical positivity (Figure 2L), and increased GFAP and Olig2 immunohistochemical positivity (Figure 2N,O). White matter, immediately below the prosencephalic lesion, is usually poorly myelinated (Figure 2P).

#### 3.1.3. Microscopic Cerebellar Analysis

In our previous publication [16], we found that prenatal vitamin C deprivation might involve PBM rupture and the sequential development of dysplastic changes in the cerebellar cortex. According to the term of intrauterine vitamin C deprivation, we established a specific histopathological scoring system (first to fourth degree of cerebellar dysplasia). To provide a more effective demonstration of the histological changes, this paper focuses on stage four of cerebellar dysplasia, specifically group E2.

Specificity of normal cortical neurogenesis in cerebellum of the control group of foetuses

In the control group at E50, we identified a well-developed external granular layer (EGL) composed of an outer, proliferative (pl) zone and an inner, premigratory (pm) zone. The first appearance of postmitotic neurons was noticed in the postmitotic zone of the EGL, and they retained strong NeuN nuclear immunoreactivity in the upcoming molecular layer (ML) and internal granular layer (IGL) (Figure 3A–C). The migratory marker doublecortin (DCX) exhibits intense neurophil staining in the pm zone of the EGL, is weak in the ML, and is again strong in the IGL (Figure 3E). Using an anti-calbindin marker, we present a characteristic unilaminar arrangement of Purkinje cell bodies (Figure 3F) with their dendritic arborisation in the molecular layer (Figure 3G). At the same time, we observed synaptogenesis, characterised by strong synaptophysin expression in ML and IGL (Figure 3D).

Specificity of aberrant cortical neurogenesis in cerebellum of vitamin C-deprived foetuses

In the fourth stage of cerebellar dysplastic changes, we noticed (as a consequence of a PBM rupture) fusion of the opposite folia with the formation of ectopic cell masses (ECM) situated in the arachnoid spaces. Despite an ectopic environment, these overmigrated neurons, which had entered the leptomeningeal spaces, showed numerous immunohistochemical evidences of their further maturation. Firstly, in the central region of those ECMs, we identified a clear postmitotic transformation of granule cells, as evidenced by increased NeuN expression (Figure 3H–J). At the same time, neurophil between ectopic cells also showed DCX expression (Figure 3L). Finally, identification of synaptophysin expression in the neurophil of EGM (Figure 3K) indicates a similar stage of granular cell development to that observed in the IGL of the control group. The degree of Purkinje cell disturbance in both experimental groups directly depends on the stage of dysplasia in the cerebellar cortex. In the first (Control group) and second stages, we observed that the cytoarchitectonic characteristics of the Purkinje cell layer (PCL) were preserved entirely. However, the third, and especially the fourth, stage of dysplasia was characterised by the loss of monoplanar orientation of Purkinje neurons, with pronounced disorganisation and aberration of the dendritic tree (Figure 3M,N).

Specificity of Cytoarchitectonic Disturbance of Bergman Glia Cells and the Process of Gliogenesis

In the control group, immunohistochemical staining for S100 clearly identified Bergmann glial cells located in the PCL (Figure 4A). From their apical part, an extension begins that crosses the ML and terminates at the pial basement membrane, creating a unique Bergmann glial-guiding scaffold for granule cells during radial migration. Fusion of opposite folia leads to a loss of glial continuity, resulting in a consequential disturbance of the polarity and spatial arrangement of Bergmann glia cells and their processes, which we identified deeply within the ECM (Figure 4F). Compared with the control group, in which nestin immunoreactivity was predominantly observed in blood vessels, the E2 group showed strong nestin expression in Bergmann glial cells and their processes (Figure 4G). The distribution of Olig2-immunolabeled oligodendroglial precursors was strictly limited to the IGL and white matter (Figure 4C), whereas in the experimental group, we also identified them in the ECM (Figure 4H). Mature form of oligodendrocytes, positive on TPPP with clear production of MBP, were limited to the zone of white matter in the control and experimental groups (Figure 4D,E,I,J).

### 3.2. Biochemical Analysis of the Amount of Hydroxyproline in Cerebellar and Cerebral Tissue Samples

The cerebrum of foetuses from the control group contains 0.31 ± 0.08 μg of hydroxyproline per mg of analysed tissue, in the E1 group 0.35 ± 0.06 μg/mg, while this value in the E2 group is 0.37 ± 0.12 μg/mg (Figure 5A). No statistically significant difference was found between the results mentioned (F(2) = 0,430, *p* = 0,657). In the cerebellum tissue samples of the control group, an average amount of 0.43 ± 0.11 μg of hydroxyproline per milligram of analysed tissue was obtained. In the other experimental group, it was 0.52 ± 0.12 μg/mg, and in the E2 group, 0.60 ± 0.11 μg/mg (Figure 5B). When comparing results across different experimental groups, no statistically significant difference was observed (F(2) = 0.176, *p* = 0.840).

### 3.3. Molecular Analysis of COL1, COL4A1 and SLC23A1 Gene Expression in Cerebral and Cerebellar Tissue Samples

Expression of COL1, COL4A1 and SLC23A1 genes in cerebral tissue: In the analysed cerebrum samples, both experimental groups showed an increase in the expression levels of the COL1 gene. However, the duration of prenatal vitamin C deficiency did not result in a statistically significant difference in the expression level of this gene (t (10) = −0.329, *p* = 0.749, d = 0.19, 95% CI [−2.64, 1.96]) (Figure 6A). The effect of the treatment on the expression of the COL4A1 gene in cerebrum tissue samples was not proven (Figure 6B). No statistically significant difference was observed between the groups (U = 14.0, *p* = 0.589, r = 0.18, 95% CI [−0.38, 0.75]). Additionally, in the vitamin C-deprived group, SLC23A1 expression was affected. Both experimental groups exhibited a significant change in expression levels, though this difference was not statistically significant (t (10) = −1.147, *p* = 0.278, d = 0.66, 95% CI [−4.61, 1.48]) (Figure 6C).

Expression of COL1, COL4A1, and SLC23A1 genes in cerebellar tissue: In the experimental groups, all three analysed genes showed treatment-dependent expression changes. Namely, the earlier withdrawal of vitamin C led to a significant increase in the expression levels of COL1, COL4A1, and SLC23A1. In contrast, a decrease in the expression levels of all analysed genes was observed in the samples from the E1 group (Figure 6D–F). Statistical analysis of Coll1 gene expression data revealed a significant difference between groups (U = 0.50, *p* = 0.001, r = 0.82, 95% CI [0.25, 1.00]). For the expression of the Coll4a1 gene, a statistically significant difference was also observed between the experimental groups (t (10) = −4.078, *p* = 0.001, d = 2.36, 95% CI [−3.45, −1.01]). Finally, the analysis of Slc23a1 expression results revealed a statistically significant difference between the two experimental groups (U = 5.0, *p* = 0.041, r = 0.60, 95% CI [0.04, 1.00]).

## 4. Discussion

One key factor determining the extent and specificity of the changes caused by prenatal vitamin C deprivation is the precise timing of its onset. Some studies indicate that vitamin deficiency in the early period leads to metabolic disorders, while in the later period, the changes are characterised by a halt in foetal growth [25,26]. The results of our study indicate that earlier and prolonged Vitamin C deprivation in the E2 group suggests a possible association with extensive foetal changes, including weight loss [25] and the development of malformations. The observed changes were diffuse, affecting osteogenesis and dentinogenesis, as well as parenchymal organ atrophy and the central nervous system. Due to the extensive nature of the study results, only neurodevelopmental changes, with an emphasis on the cerebellum, are presented here. The finding of a disorder of cerebellar foliation with lissencephaly as well as extensive prenatal cerebral haemorrhage with consequent porencephaly was the most striking. The distribution and extent of the observed cerebral haemorrhages were directly conditioned by the duration and also the moment of the onset of deprivation, which in group E2 was as early as the 10th embryonic day. In addition to fresh haemorrhage, a clear indicator that the processes may be present intrauterine was the finding of Fe ions within numerous noted hemosiderophages. Their findings indicate chronic bleeding. In the pathogenesis of fresh haemorrhage, the leading cause is vascular wall disruption. High-quality tissue fixation is achieved by perfusion, during which fixative is pumped through the circulatory system. This technical step is crucial for the appearance of fresh haemorrhage in fragile blood vessels in the experimental groups. The absence of haemorrhage in the control group indicates that these vessels were not in the same condition after all. To improve the study design, we can use immersion fixation, which reduces pressure on the blood vessels.

Similar extensive changes have been observed in experimental models based on knockout animals for the GULO [27], COL4A1/2 [28], COL4A1 [19], and SVCT2 genes [28]. Some studies indicate that a marked reduction in Vitamin C status in GULO (-/-) individuals is sufficient to cause DNA hypermethylation and consequent neural tube defects [13]. Complete mutation of the allele for COL4A1/2, which encodes both alpha chains, causes changes in individuals in the early period of development. The statistics confirmed changes were predominantly in the fragile Reichert’s membrane and capillary blood vessels [13]. On the other hand, partial mutation of only one collagen chain, COL4A1, resulted in prolonged foetal survival and the development of later intrauterine cerebral haemorrhage, leading to porencephaly, a finding identical to that in our study [29]. Similar changes in the type of extensive haemorrhage in the cerebrum and brainstem, accompanied by neuronal loss with pronounced apoptosis, were observed in mice with a mutation in the SVCT2 gene, which encodes the sodium-dependent vitamin C transporter [28].

It is important to note that significant changes in the type of subarachnoid haemorrhage resulting in porencephaly were observed exclusively in the E2 group. Most likely, we noted that the earlier onset and longer duration of intrauterine deprivation are associated with this finding. The brain defect wall consisted of reduced acidophilic neurons, foamy macrophages, hemosiderophages, glial cells, and capillaries. The changes described corresponded to areas of subacute and chronic infarction [30]. Although subarachnoid haemorrhage is a common finding at autopsy in premature foetuses, the aetiology is still unclear, and hypoxia, capillary fragility, coagulopathy, and sepsis have been suggested as initiating factors [31].

Histological analysis of the cerebrum predominantly revealed endothelial dysfunction, as supported by numerous studies demonstrating disturbances in the expression of collagen type IV, laminin, and elastin [32]. However, we observed indirect changes in the PBM, characterised by a decrease in the number of neurons in the surrounding areas. This is confirmed by studies on COL4A1/2 and laminin γ1 knock-out individuals, in which protrusions of neuroembryonic tissue into the surrounding mesenchyme can be clearly observed [13,14,33]. In contrast, the neurogenesis and the delicate foliation process in the cerebellum [34]. This results in distinct neuropathological changes compared to those observed in the cerebrum. The flattening of the brain’s surface observed in the E2 group may be associated with extensive cerebellar cortical dysplasia.

A previous study [25] detailed the specific alterations that occur based on whether deprivation starts on the 10th or 20th intrauterine day. By examining a midsagittal section of the vermal region of the cerebellum, it becomes possible to not only identify dysplastic changes but also assess their degree and distribution. While there are no studies on guinea pigs, research involving knock-out mice for the genes encoding the β1-class integrin [16] and the α1 subunit of laminin [32] provides data on disorders related to cerebellar foliation and fissuration.

This study outlined the specifics of dysplastic areas concerning neurogenesis and gliogenesis. The defect in the PBM and the fusion of the adjacent folia may be related to EGL cells overmigrating into the subarachnoid space. Although in an ectopic environment, granule cells exhibited postmitotic neuronal characteristics, with clear signs of synaptogenesis. This phenomenon has also been described in other studies, with confirmation provided by light microscopy and transmission and scanning electron microscopy [35,36]. Disturbance in the development of EGL and IGL led to loss of polarity and Purkinje cell arborization, as reported in other studies [16].

In humans, neuropathological changes resulting from PBM rupture are associated with Lissencephaly type II. Macroscopically, it is characterised by the absence of gyri and folia, with a flattened brain surface. Histologically, evidence of cerebral and cerebellar dysplasia is observed [37]. Previous studies have mainly attributed the pathogenesis of this disorder to genetic mutations, particularly in the POMT1, POMT2, POMGNT1, LARGE, FKRP, and FKTN genes [38,39,40,41]. However, in over one-third of cases, a genetic link to the onset of the disease remains unidentified [42].

Hydroxyproline is used to quantify collagen by measuring its concentration in a sample, as it is a unique amino acid predominantly found in collagen [42]. Numerous studies have established a connection between vitamin C deficiency and disturbance in hydroxyproline formation. Most of them indicate a decrease in hydroxyproline in the bone matrix [43,44], connective tissue of blood vessels [45], the uterus [46], and the skin of guinea pigs [43,44] in a state of deprivation. Additionally, studies on Gulo-/- mice have shown that vitamin C deficiency affects the amount of hydroxyproline in the skin and the prolactin-stimulated mammary gland. Namely, Gulo-/- individuals had statistically significantly higher hydroxyproline levels than individuals in the control group [47]. The results of our study on brain tissue did not show statistically significant differences, most likely due to the low collagen content in the analysed tissues. Although colourimetric analysis is as sensitive as high-performance liquid chromatography (HPLC), more sensitive methods such as liquid chromatography–mass spectrometry (LC-MS) may be a better choice for brain tissue [48]. However, the specificity of the organ itself, as well as the complexity of the scurvy process, indeed dictates whether a decrease or increase in the amount of this biochemical gradient will be detected [49]. Also, non-significant differences may reflect insufficient sensitivity of the colourimetric assay rather than the method used, for example, the HPLC-MS method.

The decrease in the expression of genes for collagen type IV and elastin in vitamin C deficiency also confirmed the close connection between vitamin C and tissue-building elements. All this was accompanied by a decrease in the amounts of the proteins mentioned above isolated from blood vessels at that time [28]. The results of our study show a statistically significant reduction in the expression of the COL1 and COL4A genes in cerebellar tissue samples from the E1 group. The complexity of analysing and interpreting gene expression and post-translational changes is further confirmed by a study showing that the application of Vitamin C to isolated aortic tissue increases mRNA expression of collagen 1 and decreases elastin [50].

The transport of vitamin C into cells depends on the sodium-dependent vitamin C transporters, number 1 (SVCT1) and 2 (SVCT2), which are encoded by the genes SLC23A1 and SLC23A2, respectively. Thanks to them, the concentration of vitamin C can be several tens of times higher inside the cell [51,52]. Given the variability in Vitamin C concentration among tissues [53], the literature reports conflicting data on the expression of the genes mentioned above. Namely, in a study of vitamin C deprivation in guinea pigs, no changes in SVCT expression were observed in the kidney and brain tissue [54]. On the other hand, in the cartilage tissue of guinea pigs experiencing vitamin C deprivation, an increase in gene expression for SVCT2 was observed, but not for SVCT1 [55]. In the condition of chronic pre- and postnatal deprivation in guinea pigs, an increase in SVCT2 expression was also not observed [56].

In addition to its importance in collagen biochemistry, vitamin C has strong antioxidant properties that neutralise free radicals [57]. At the same time, the regeneration of the oxidised form of vitamin E (a lipid antioxidant) to its active form also affects antioxidant activity. Applied alone [58] or combined with vitamin E [59], Vitamin C demonstrated a practical benefit of antioxidant potential through reducing oxidative stress-induced toxicity in embryos, as shown by improved blastocyst development.

Based on all of the above, the changes caused by Vitamin C deficiency in the prenatal period certainly reflect complex biochemical dysfunction.

Despite the compelling results, this study had limitations and can undoubtedly be improved by the following: volumetric and stereological brain analysis; Vitamin C serum and tissue level determination in both mothers and foetuses; quantification of specific types of collagens in tissues using the Western blot or other methods; and extending the investigation beyond prenatal deprivation to examine its consequences during the postnatal period of life.

## 5. Conclusions

These findings represent preliminary associations between vitamin C deficiency and associated morphological, biochemical, and molecular changes, but require further validation. The observed changes may offer a new perspective on using this animal model to study potential human disease conditions relevant to public health, gynaecology, and obstetrics research.

## Figures and Tables

**Figure 1 nutrients-17-03484-f001:**
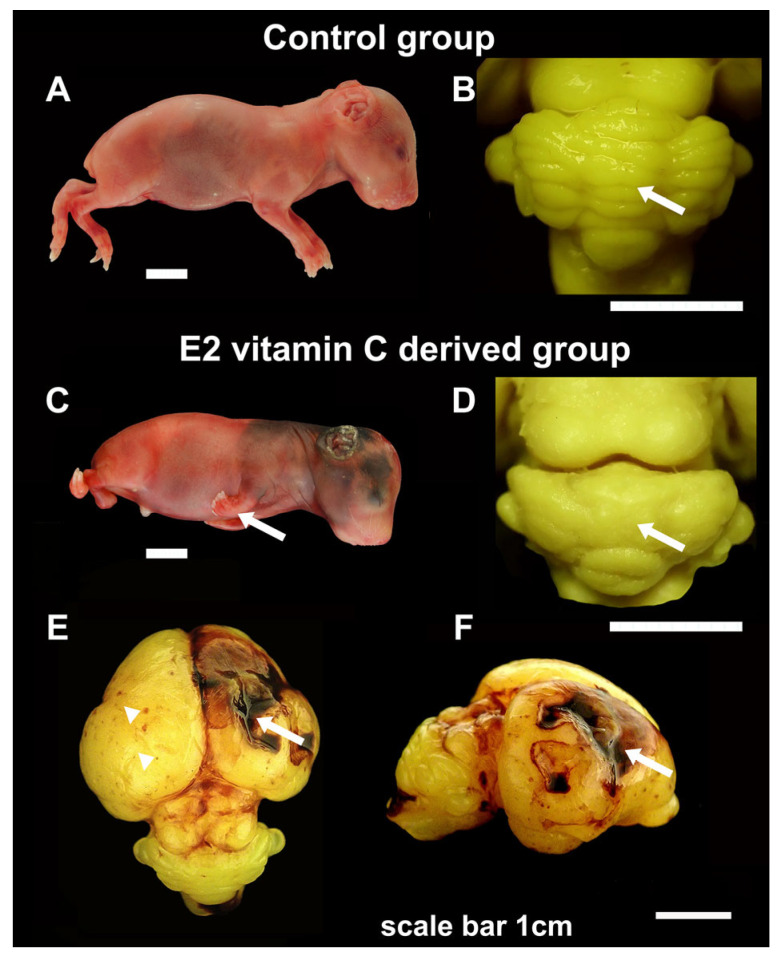
Gross characteristics of foetuses, cerebellum and cerebrum in the Control and E2 vitamin C-deprived group. Macroscopic view of control (**A**) and a vitamin C-deprived foetus with clubfoot deformity (arrow, (**C**)). Dorsal view of cerebellum shows regular arrangement in the Control group (arrow, (**B**)) and the absence of folia in cerebellar hemispheres and vermis on the cerebellar surface (arrow, (**D**)) in the E2 group. The cerebrum surface of the Control group in E50 showed a smooth surface. In contrast, in the E2 group, the noted changes ranged from focal petechiae (arrow, (**E**)) to extensive intraparenchymal bleeding with consequential porencephalic defects (arrowhead, (**E**,**F**)).

**Figure 2 nutrients-17-03484-f002:**
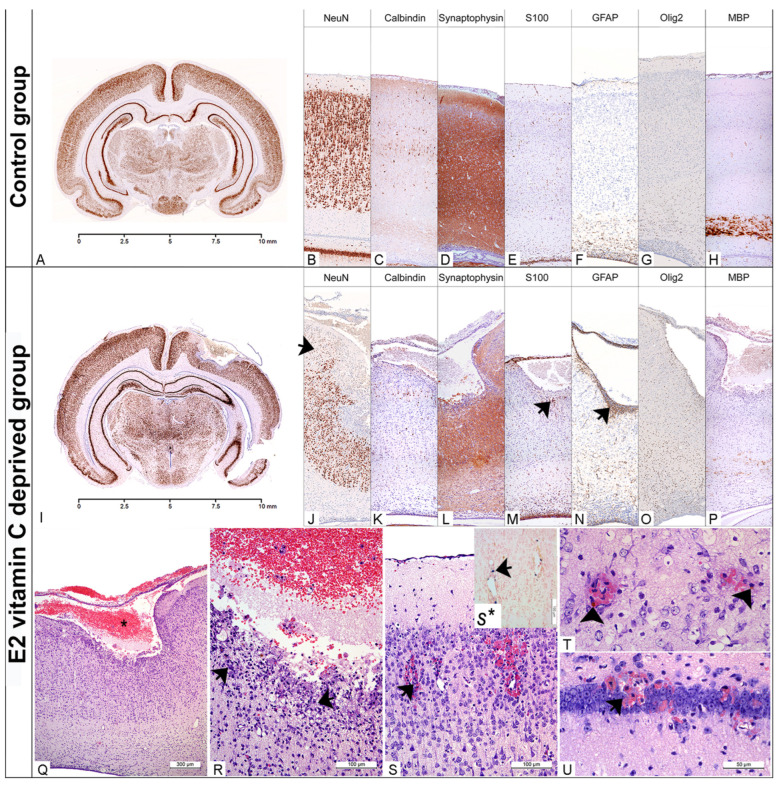
Neuropathological findings of the cerebral region in the Control and the E2 vitamin C-deprived group. Low magnification of the coronal section of the preserved cerebrum in control (**A**) and an extensive porencephalic region in the E2 group (**I**). NeuN and Calbindin exhibit a characteristic laminar organisation of the cerebral cortex in control (**B**,**C**) and loss of lamins (od I to IV), as well as a reduction in neurons (arrow, (**J**)) in bleeding-affected regions in the E2 group (**J**,**K**). An appropriate number of GFAP-positive glial cells (**E**), preserved synaptic activity (**D**) and rare S100-positive histiocytes (**F**) are detected in the Control group. The encephalomalatic wall in porencephaly (asterisks, (**Q**)) showed the presence of degenerated neurons (arrow, (**R**)), foamy histiocytes stained with S100 (arrow, (**M**)), reactive astrocytes marked by GFAP (arrow, (**N**)), and a reduction in synaptic activity (arrow, (**L**)). The increased number of Olig2-positive oligodendrocytes in E2 (**O**) compared to the control group (**G**) led to a consequential absence of myelination in E2 (**P**) compared to the control group (**H**). In the rare animals that belonged to the E1 group, and in all from the E2 group, histological signs of fresh (arrow (**S**–**U**)) and old bleeding (hemosiderophages, arrowhead, (**T**)) with Perl’s positive Fe^2+^ were detected (arrow (**S***)).

**Figure 3 nutrients-17-03484-f003:**
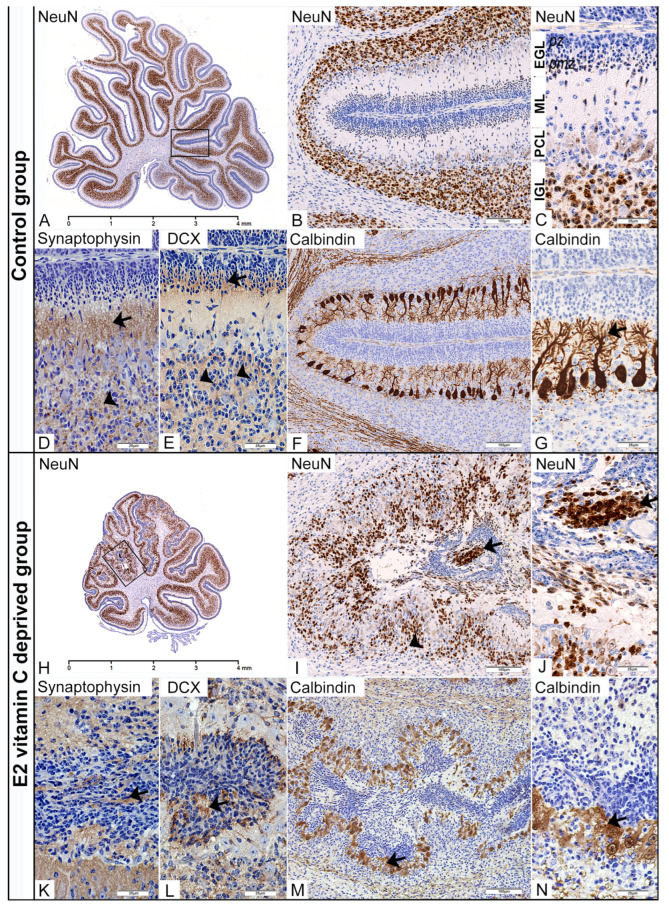
Immunohistochemical characteristics of the cerebellar corticogenesis in the Control and the E2 vitamin C-deprived group. EGL—external granular layer; pz—proliferative zone; pmz—premigratory zone; ML—molecular layer; PCL—Purkinje cell layer; IGL—internal granular layer. Low magnification of the midsagittal section of the vermis with visible normal (**A**) and dysplastic corticogenesis (**H**) presented in the black frame area. High magnification of the mentioned area in the control group shows regular distribution of NeuN (**B**,**C**) and DCX positivity (**E**) of postmitotic neurons in the pmz (arrow) and IGL (arrowhead). Additionally, in the control group, an unilaminar arrangement of Purkinje cell bodies (**F**) with their dendritic arborization in the ML (arrow, (**G**)) and vigorous synaptic activity in the ML (arrow, (**D**)) and IGL (arrowhead, (**D**)) was observed. The same black frame in the E2 group shows intensive disturbance with ectopic distribution of NeuN (arrow, (**I**,**J**)) and DCX neuronal positivity (**L**) in the subarachnoid space (arrow) and an intensive decrease in numerical density in the IGL (arrowhead, (**I**)). At the same time, PCL is associated with a loss of monoplanar orientation of Purkinje cell bodies, accompanied by a disturbance of the dendritic tree (arrow, (**M**,**N**)). Synaptic activity was also disturbed with an unusual presence in overmigrated neurons (arrow, (**K**)).

**Figure 4 nutrients-17-03484-f004:**
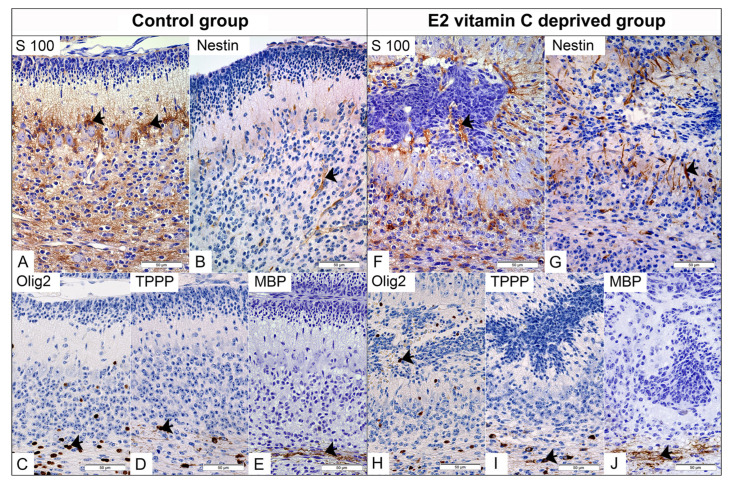
Immunohistochemical characteristics of the cerebellar gliogenesis in the Control and the E2 vitamin C-deprived group. S100 positive Bergmann glial in PCL of Control (arrow, (**A**)) and in PCL and inside subarachnoid ectopic neuronal masses of E2 group (arrow, (**F**)). Expected selective Nestin staining of foetal vascular endothelium (arrow) in the control group (**B**). An unexpected additional Bergman glia (arrow) is stained for Nestin in the E2 group (**G**). The normal distribution of Olig2-positive immature oligodendrocytes is observed in the PCL and IGL of the control group (arrow, (**C**)). In contrast, the E2 group exhibits an atypical finding: Olig2-positive cells in ectopic masses (arrow, (**H**)). On the other hand, TPPP-positive mature oligodendrocytes (arrow) are present in the white matter regions of both the control group (**D**) and the E2 group (**I**). Both groups show clear signs of myelination, as evidenced by MBP positivity (arrow) in the control group (**E**) and the E2 group (**J**).

**Figure 5 nutrients-17-03484-f005:**
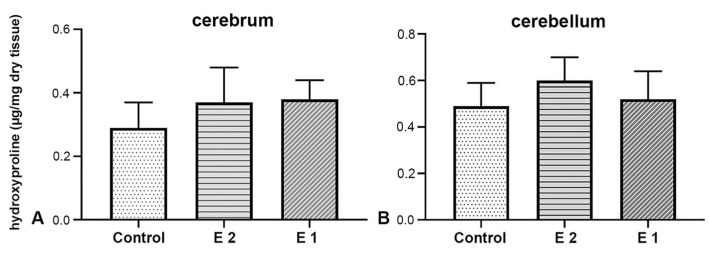
Biochemical analysis of hydroxyproline concentration in cerebrum (**A**) and cerebellum (**B**) tissue samples.

**Figure 6 nutrients-17-03484-f006:**
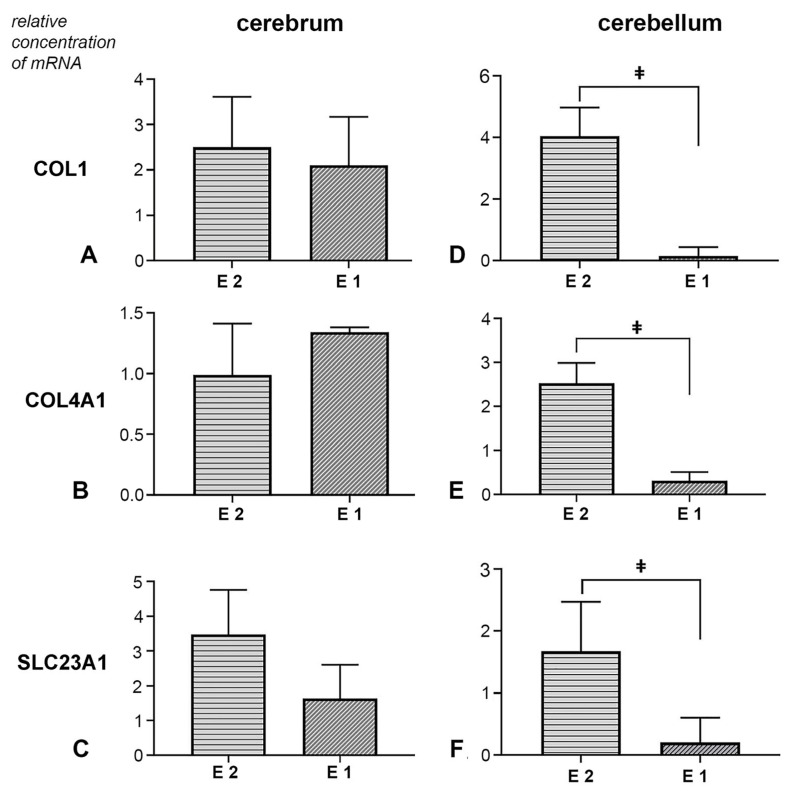
Effect of prenatal vitamin C deprivation on the mRNA expression. Analysed cerebral and cerebellar tissue for COL1 (**A**,**D**), COL4A1 (**B**,**E**) and SLC23A1 gene expression (**C**,**F**). ǂ Statistically significant difference between E1 and E2 groups. A value of *p* < 0.05 was considered statistically significant.

**Table 1 nutrients-17-03484-t001:** Clonality, source and dilutions of antibodies used in immunohistochemistry.

Antibody Name	Clone	Source	Company	Catalogue No.	Dilution
Anti-NeuN	EPR12763	Rabbit monoclonal	Abcam (Cambridge, UK)	ab177487	1:3000
Anti-DCX	polyclonal	Rabbit polyclonal	Abcam (Cambridge, UK)	ab18723	1:500
Anti-Nestin	EPR1301(2)	Rabbit monoclonal	Abcam (Cambridge, UK)	ab176571	1:250
Anti-S100	EP1576Y	Rabbit monoclonal	Abcam (Cambridge, UK)	ab52642	1:1000
Anti-MBP	12	Rat monoclonal	Abcam (Cambridge, UK)	ab7349	1:100
Anti-Synaptophysin	SYP02	Mouse monoclonal	Thermo Scientific (Waltham, MA, USA)	MS-1150-S0	1:40
Anti-GFAP	polyclonal	Rabbit polyclonal	Thermo Scientific (Waltham, MA, USA)	RB-087-A0	1:200
Anti Calbindin D-28k	CB38	Rabbit monoclonal	Swant (Tägerwilen, Switzerland)	-	1:5000
Anti-Olig2	polyclonal	Rabbit polyclonal	IBL (Fujioka, Japan)	18953	1:100
Anti-TPPP	6C10	Mouse monoclonal	Flow Labs UK (London, UK)	-	1:2000

**Table 2 nutrients-17-03484-t002:** Primer sequences of housekeeping and target genes.

Gene	Primers	Primer Length
COL1	F: 5′-ATGTCTAGGGTCTAGACATGTTCA-3′	24 bp
	R: 5′-CCTTGCCGTTGTCGCAGACG-3′	20 bp
COL4A1	F: 5′-TATCTCTGGGGACAACATCCG-3′	21 bp
	R: 5′-CATCTCGCTTCTCTCTATGGTG-3′	22 bp
SLC23A1	F: 5′-TACCTGACATGCTTCAGTGG-3′	20 bp
	R: 5′-CGGCTGCCCACCTTGGTAAT-3′	20 bp
GAPDH	F: 5′-GCGCCGAGTATGTAGTGGAA-3′	20 bp
	R: 5′-TGATTCACGCCCATCACGAA-3′	20 bp

Note: Primer design was performed using the Primer Express programme, and exact gene sequences were obtained from the NCBI database.

**Table 3 nutrients-17-03484-t003:** The distribution of intracranial haemorrhage.

		Intra-Axial Haemorrhage (Intraparenchymal)				Extra-Axial Haemorrhage (Subarachnoid)
Group No. of Foetuses		Cerebral Cortex	Hippocampus	Thalamus and Hypothalamus	Cerebellum	
Control	20	0	0	0	0	0
E1	21	7 ^1^	6 ^4^	7 ^7^	2 ^10^	0
E2	19	19 ^2,3^	19 ^5,6^	19 ^8,9^	19 ^11,12^	11 ^13,14^

^1^ Con vs. E1 *p* < 0.01; ^2^ Con vs. E2 *p* < 0.001; ^3^ E1 vs. E2 *p* < 0.001; ^4^ Con vs. E1 *p* < 0.01; ^5^ Con vs. E2 *p* < 0.001; ^6^ E1 vs. E2 *p* < 0.001; ^7^ Con vs. E1 *p* < 0.01; ^8^ Con vs. E2 *p* < 0.001; ^9^ E1 vs. E2 *p* < 0.001; ^10^ Con vs. E1 *p* < 0.01; ^11^ Con vs. E2 *p* < 0.001; ^12^ E1 vs. E2 *p* < 0.001; ^13^ Con vs. E2 *p* < 0.001; ^14^ E1 vs. E2 *p* < 0.001.

## Data Availability

The original contributions presented in this study are included in the article/Appendix A. Further inquiries can be directed to the corresponding author.

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
