# Peer review of "Neurodevelopmental Changes in the Guinea Pig Brain Caused by Time-Limited Complete Vitamin C Deprivation"

_nutrients, 2025, doi:10.3390/nu17213484_

Round 1

Reviewer 1 Report

Comments and Suggestions for Authors

Neurodevelopmental changes in the guinea pig brain caused by time-limited complete vitamin C deprivation

The manuscript is globally interesting and well-written. As a general remark, I have appreciated it. You can find my appraisal, section by section, with suggestions, comments, and questions, as follows:

Introduction: The introduction is concise, synthetic. Despite this, I have some specific suggestions: In the first two paragraphs,  the link between deficiency in the vitamin C synthesis, which is a physiological process common to several species, is related to pathological conditions, that is, bleeding from small blood vessels. A deficiency in vitamin C needs to be introduced in a better way. Please describe this link better. Similarly, the link between PBM and its development and preservation needs to be stated better.

Despite this, the remaining paragraphs of the introduction, including the aims, are well-written and clear.

Methods: The methods are described in detail, and no issues are detected. Despite this, the statistical analyses that you described in the sections must be stated better. I agree with the application of the ANOVA. However, you stated: The Mann-Whitney U test was employed for the comparison of changes

between two groups. The Mann-Whitney test is a non-parametric inferential test. In this case, you must check the prerequisites for the application of a parametric test (normality, homoscedasticity, skewness, etc.) and then, in the case of their violation, you can apply a non-parametric. Violating the application of Parametric and Non-Parametric Tests in a mixed manner can lead to invalid test results. In the first case, the results cannot be tested based on the F distribution (ANOVA). Please, fix it.

Results: 3.1. Morphological analysis of cerebellar and cerebral tissue: The description is intriguing and extremely detailed. In Table 3, I suggest applying the chi-squared test, reporting the values, df, and p-values. 3.2. Biochemical analysis of the amount of hydroxyproline in cerebellar and cerebral tissue samples: About the content of this section, you need to add the value of the statistical test (please refer to my previous concern), F or(df) =xx, and p values. If the F cannot be calculated, I suggest applying the KW test.

Discussion: The discussion is in line with the results 3.1, and it is interesting. Despite this, since I do not agree with the statistical analysis, I suggest modifying the discussion accordingly. Moreover, the limitations need to be clearly stated in the discussion, as well as the future directions.

Author Response

Dear reviewer,

We thank you and the reviewer for commenting on our manuscript, identifying its weakness, and, moreover, providing us the opportunity to strengthen our research and raise its quality.

Comments 1: Introduction: The introduction is concise, synthetic. Despite this, I have some specific suggestions: In the first two paragraphs,  the link between deficiency in the vitamin C synthesis, which is a physiological process common to several species, is related to pathological conditions, that is, bleeding from small blood vessels. A deficiency in vitamin C needs to be introduced in a better way. Please describe this link better. Similarly, the link between PBM and its development and preservation needs to be stated better. Despite this, the remaining paragraphs of the introduction, including the aims, are well-written and clear.

Response 1 :

We made an effort to provide information on the clinical significance and current problems associated with scurvy. Also, we extend the link to the preservation of PBM and Vitamin C.

  • in the new version of the introduction, we added:

.... collagen synthesis [4]. Clearly explained pathomechanism represents a long-forgotten clinical entity called scurvy. However, scurvy is again on the rise in developed countries, mainly due to poor diets high in processed foods and low in fresh fruits and vegetables. At-risk groups include alcoholics, smokers, pregnant women, individuals with eating disorders like anorexia, the elderly, and patients with chronic gastrointestinal diseases such as Crohn's disease. Smoking notably reduces vitamin C absorption and increases the body’s needs.

… of the basement membrane [5,6]. The pial basement membrane (PBM) serves as an essential scaffold for cerebral cortex development. Composed of the pia mater and the glia limitans, it provides a pathway for migrating neuroblasts and organises radial glia fibres.

 Comments 2:  Methods: The methods are described in detail, and no issues are detected. Despite this, the statistical analyses that you described in the sections must be stated better. I agree with the application of the ANOVA. However, you stated: The Mann-Whitney U test was employed for the comparison of changes between two groups. The Mann-Whitney test is a non-parametric inferential test. In this case, you must check the prerequisites for the application of a parametric test (normality, homoscedasticity, skewness, etc.) and then, in the case of their violation, you can apply a non-parametric. Violating the application of Parametric and Non-Parametric Tests in a mixed manner can lead to invalid test results. In the first case, the results cannot be tested based on the F distribution (ANOVA). Please, fix it.

Response 2:

We accept your suggestion completely and the improved methodology.

  • In the new version of the text, we added:

2.8. Statistical Analysis

Statistical analysis was performed using IBM SPSS statistical software, version 26.0 (IBM Corp., Armonk, NY, USA). Data were reported as the mean ± standard deviation (SD) or standard error of the mean (SEM). The normality of continuous variable distributions was assessed using the Kolmogorov-Smirnov and Shapiro-Wilk tests. In addition, the choice of alternative methods (nonparametric techniques) was based on variable type, coefficient of variance, and the results of the Levene test for homogeneity of variances. Because the assumptions for the chi-square test were not fully met, Fisher’s exact test (Monte Carlo method, two-sided) was applied to obtain precise significance values for differences in the frequency of intracranial haemorrhages among the experimental groups. A one-way analysis of variance (ANOVA) was used to compare groups for hydroxyproline concentrations. Post hoc testing for ANOVA was performed using Tukey’s test. For the analysis of mRNA relative concentration, an independent-samples T-test or its nonparametric alternative, the Mann–Whitney U test, was used. For each test, the corresponding effect size was calculated (Cohen’s d for t-tests, r for non-parametric tests). Where applicable, 95% confidence intervals (CIs) for the effect sizes were also reported to indicate the precision of the estimates. The difference between groups was considered statistically significant for a p-value less than 0.05 (p < 0.05).

 Results:

Comments 3:  Morphological analysis of cerebellar and cerebral tissue: The description is intriguing and extremely detailed. In Table 3, I suggest applying the chi-squared test, reporting the values, df, and p-values.

Response 3:

We accept your suggestion completely and the improved results.

  • In the new version of the text, we added:
  • Table 3. The distribution of intracranial haemorrhage

Group                   No.of fetuses

intra-axial haemorrhage (intraparenchymal)

extra-axial hemorrhage (subarachnoid)

cerebral cortex

hippocampus

thalamus and hypothalamus

cerebellum

Control

E1

E2

20

21

19

0

71

192,3

0

64

195,6

0

77

198,9

0

210

1911,12

0

0

1113,14

1 Con vs E1 p < 0,01; 2 Con vs E2 p < 0,001; 3 E1 vs E2 p < 0,001;  4 Con vs E1 p < 0,01; 5 Con vs E2 p < 0,001; 6 E1 vs E2 p < 0,001; 7 Con vs E1 p < 0,01; 8 Con vs E2 p < 0,001; 9 E1 vs E2 p < 0,001; 10 Con vs E1 p < 0,01; 11 Con vs E2 p < 0,001; 12 E1 vs E2 p < 0,001; 13Con vs E2 p < 0,001; 14 E1 vs E2 p < 0,001;

Statistically significant differences in the occurrence of hemorrhages among the groups were observed in the cerebral cortex (χ²(2) = 40.995, p < 0.001; Fisher’s exact test, Monte Carlo method), hippocampus (χ²(2) = 42.367, p < 0.001), thalamus and hypothalamus (χ²(2) = 40.995, p < 0.001), cerebellum (χ²(2) = 52.046, p < 0.001), and in the case of extra-axial (subarachnoid) hemorrhage (χ²(2) = 29.066, p < 0.001).

Comments 4: Biochemical analysis of the amount of hydroxyproline in cerebellar and cerebral tissue samples: About the content of this section, you need to add the value of the statistical test (please refer to my previous concern), F or(df) =xx, and p values. If the F cannot be calculated, I suggest applying the KW test.

Response 4:

We accept your suggestion completely and the improved results.

  • In the new version of the text, we added:

3.2. Biochemical analysis of the amount of hydroxyproline in cerebellar and cerebral tissue samples

The cerebrum of fetuses from the control group contains 0.31±0.08 μg of hydroxyproline per mg of analysed tissue, in the E1 group 0.35±0.06 μg/mg, while this value in the E2 group is 0.37±0.12 μg/mg (Figure 5A). No statistically significant difference was found between the results mentioned (F(2) = 0,430, p = 0,657). In the cerebellum tissue samples of the control group, an average amount of 0.43±0.11 μg of hydroxyproline per milligram of analysed tissue was obtained. In the other experimental group, it was 0.52±0.12 μg/mg, and in the E2 group, 0.60±0.11 μg/mg (Figure 5B). When comparing results across different experimental groups, no statistically significant difference was observed (F(2) = 0.176, p = 0.840).

 Comments 5: Discussion: The discussion is in line with the results 3.1, and it is interesting. Despite this, since I do not agree with the statistical analysis, I suggest modifying the discussion accordingly. Moreover, the limitations need to be clearly stated in the discussion, as well as the future directions.

Response 5:

  • In the new version of the text, we added:

… an increase in SVCT2 expression was also not observed [55].

Despite the compelling results, particularly in the histological analysis, the study had limitation and can undoubtedly be improved by: analyzing serum and tissue concentrations of Vitamin C levels in both mothers and fetuses to determine biological availability; quantifying specific types of collagen in tissues using the western blot method to assess the collagen profile; and extending the investigation beyond prenatal deprivation to examine its consequences during the postnatal period of life.

Reviewer 2 Report

Comments and Suggestions for Authors

This manuscript by Ivan Capo investigated neurodevelopmental effects of time-limited, complete prenatal vitamin C deprivation in guinea pigs, focusing on macroscopic and microscopic brain changes, hydroxyproline content, and mRNA expression of Col1, Col4a1, Slc23a1. The topic fits Nutrients and leverages a species with endogenous vitamin C synthesis loss (GULO), giving translational relevance for scurvy-related neurodevelopmental pathology. The study presents striking neuropathology in the early-deprived group (E2), including subarachnoid and intraparenchymal hemorrhages, porencephaly, and cerebellar dysplasia, and documents altered gene expression in cerebellum. Overall, the work is potentially impactful, but the Introduction, design transparency, quantification rigor, and statistical reporting require substantial strengthening before the conclusions can be supported.

In my opinion, therefore, this manuscript is not recommended for publication in its present form.

Major points

  1. Introduction:

The Introduction is over-segmented and somewhat repetitive, diluting the core rationale. Much space reiterates general facts (loss of GULO, collagen/ascorbate biochemistry, PBM importance) without crisply mapping the knowledge gap and specific hypotheses that this experiment tests. I recommend condensing to 3–4 paragraphs. This will improve readability and better justify the selected endpoints Discussion (serotonin pathway)

  1. Study design

Attrition/selection: In E2, many dams failed to maintain pregnancy and were replaced. Please present a CONSORT-like flow of dams and fetuses (per group: enrolled, excluded with reasons, analyzed), and discuss attrition bias. Clarify whether replacements were pre-specified and how litter effects were handled.

Exposure verification: Provide maternal/fetal ascorbate concentrations (plasma and, ideally, brain) at E50 to confirm deprivation severity per group; gene expression alone is insufficient to validate exposure. If archived samples are unavailable, acknowledge this as a limitation.

Outcomes hierarchy: Pre-specify primary (e.g., hemorrhage incidence/extent, defined dysplasia score) vs secondary outcomes (hydroxyproline, mRNA), and state whether volume measurements (cerebrum/cerebellum) or stereology were planned. The text notes smaller cerebrum volume in E2 but says no volumetry was conducted; either quantify or avoid causal language.

  1. Methods and measurements:

Pathology quantification: The neuropathology is compelling, but largely qualitative. Please add quantitative metrics: hemorrhage incidence per region, lesion area/volume (image analysis), dysplasia grade distribution, and myelination indices (MBP-positive area). Provide inter-rater agreement and blinding details.

Hydroxyproline assay: Given the low collagen content of brain, discuss assay sensitivity, LOD/LOQ, and normalization. I wonder that Non-significant differences may reflect insufficient sensitivity rather than biology.

Molecular endpoints: The Figure 6 legend shows “COL2” in places where the text focuses on Col4a1—please correct any nomenclature inconsistencies and ensure primer targets match reported genes.

  1. Statistics and reporting:

State biological n at the dam and fetus levels (accounting for litter as a random effect), not only the number of fetuses/sections. If multiple fetuses per litter were included, use models that adjust for clustering.

Unify the analysis plan: specify normality/variance checks, ANOVA vs. non-parametric criteria, post-hoc corrections, and report effect sizes with 95% CIs alongside P values. Clarify when Mann–Whitney vs ANOVA was chosen, and apply a consistent multiple-comparison strategy.

Figures should show individual data points, define what bars/points represent, and indicate blinding for image analyses.

  1. Interpretation and discussion

    The proposed link from vitamin C deprivation to PBM rupture leading to dysplasia and hemorrhage is plausible and consistent with prior models. However, causal inference would be stronger with quantification of PBM components (e.g., COL4A1, laminin) and objective hemorrhage metrics. Where such data are unavailable, please temper causal language and acknowledge perfusion–fixation as a possible contributor.

Please also distinguish clearly what replicates or extends prior work versus what is novel (e.g., the strict timing of complete deprivation and the staging of cerebellar dysplasia).

I recommend that conclusions regarding hydroxyproline be appropriately qualified, given the assay’s constraints in brain tissue.

I hope these comments will be helpful.

Comments on the Quality of English Language

Overall readability is adequate, but the manuscript needs light-to-moderate language editing. Recurring issues include: inconsistent terminology (e.g., PBM vs. BM; ROS used without definition), gene/protein nomenclature drift (COL4A1 vs. Col4a1; SLC23A1 vs. Slc23a1), article and preposition errors (“the” overuse; “effect on/of” mix-ups), tense inconsistency between Methods/Results, and punctuation/spelling slips (e.g., H2DCFDA vs. H2DCFDA/DCF; “perfusion–fixation” hyphenation; figure/axis labels with capitalization mismatches). Figure legends occasionally omit statistical details or units. A professional copy-edit to standardize style, fix typos, and harmonize nomenclature would be beneficial.

Author Response

Respons to reviewer

Dear reviewer,

We thank you and the reviewer for commenting on our manuscript, identifying its weakness, and, moreover, providing us the opportunity to strengthen our research and raise its quality.

Reviewer 2

Major points

Comments 1: Introduction: The Introduction is over-segmented and somewhat repetitive, diluting the core rationale. Much space reiterates general facts (loss of GULO, collagen/ascorbate biochemistry, PBM importance) without crisply mapping the knowledge gap and specific hypotheses that this experiment tests. I recommend condensing to 3–4 paragraphs. This will improve readability and better justify the selected endpoints Discussion (serotonin pathway)

Response 1:

To improve the introduction's readability, we have condensed it to 4 paragraphs, including the objective. We have also added clinical aspects of scurvy and updated information on new critical points for its development.

In the new version, we added:

…of the basement membrane [5,6]. Clearly explained pathomechanism represents a long-forgotten clinical entity called scurvy. However, scurvy is again on the rise in developed countries, mainly due to poor diets high in processed foods and low in fresh fruits and vegetables. At-risk groups include alcoholics, smokers, individuals with eating disorders like anorexia, the elderly, and patients with chronic gastrointestinal diseases such as Crohn's disease. Smoking notably reduces vitamin C absorption and increases the body’s needs. However, all available studies linking vitamin C deficiency and brain disorders are mainly on experimental animals.

The pial basement membrane (PBM) serves as an essential scaffold for cerebral cortex development. Composed of the pia mater and the glia limitans, it provides a pathway for migrating neuroblasts and organises radial glia fibres. During brain development, the preservation and integrity of the pial basal membrane (PBM) are key factors for proper neuronal migration [7,8]. …

Study design

Comments 2: Attrition/selection: In E2, many dams failed to maintain pregnancy and were replaced. Please present a CONSORT-like flow of dams and fetuses (per group: enrolled, excluded with reasons, analyzed), and discuss attrition bias. Clarify whether replacements were pre-specified and how litter effects were handled.

Response 2:

The time-dependent onset of deprivation was key to demonstrating the importance of Vitamin C. Accordingly, earlier deprivation resulted in a higher number of abortions in the E2 group. Accordingly, at your suggestion, we expanded on and completed this key moment in the study's design.

In the new version, we added: … two experimental groups, E1 (n = 10) and E2 (n = 10). Experimental animals that could not achieve or maintain pregnancy during the experiment (0 in the Control group, 1 in the E1 group, and 9 in the E2 group) were excluded from the study and replaced with randomly selected new dams. A clear sign that an abortion had occurred in the dam (before E50) was the appearance of blood in the vaginal opening and sudden weight loss. Final flow of dams was: control group, enrolled (n = 0), excluded with reasons (n = 0), analyzed (n = 10); E1 group, enrolled (n = 11), excluded with reasons (n = 1), analyzed (n = 10); and in E2 group, enrolled (n = 19), excluded with reasons (n = 9), analyzed (n = 10);

Comments 3: Exposure verification: Provide maternal/fetal ascorbate concentrations (plasma and, ideally, brain) at E50 to confirm deprivation severity per group; gene expression alone is insufficient to validate exposure. If archived samples are unavailable, acknowledge this as a limitation.

Response 3:

Analysis of vitamin C concentration in serum or tissue was not performed. This limitation is noted in the discussion, which highlights the relevance of this analysis for enhancing the study.

In the new version of the discussion, we added:

… an increase in SVCT2 expression was also not observed [55].

Despite the compelling results, the study had limitations and can undoubtedly be improved by: volumetric and stereological brain analysis; Vitamin C serum and tissue level determination in both mothers and fetuses; quantification of specific types of collagens in tissues using the western blot or other methods; and extending the investigation beyond prenatal deprivation to examine its consequences during the postnatal period of life.

Comments 4: Outcomes hierarchy: Pre-specify primary (e.g., hemorrhage incidence/extent, defined dysplasia score) vs secondary outcomes (hydroxyproline, mRNA), and state whether volume measurements (cerebrum/cerebellum) or stereology were planned. The text notes smaller cerebrum volume in E2 but says no volumetry was conducted; either quantify or avoid causal language.

Response 4:

In methodology (study design) we pre-specify analysing of primary (e.g., hemorrhage incidence/extent, defined dysplasia score) vs secondary outcomes (hydroxyproline, mRNA):

…, and E2 had 31. For histological analysis, we use 20 from the control group, 21 from E1, and 19 from E2; for biochemical and molecular analysis, we use six fetal brains from each group.

To avoid causal language, we removed the entire sentence: Overall, the cerebrum of the E2 group was somewhat smaller in volume than that of the control and E1 groups; however, no detailed volume analysis was conducted.

Methods and measurements:

Comments 5: Pathology quantification: The neuropathology is compelling, but largely qualitative. Please add quantitative metrics: haemorrhage incidence per region, lesion area/volume (image analysis), dysplasia grade distribution, and myelination indices (MBP-positive area). Provide inter-rater agreement and blinding details.

Response 5:

  • As we mentioned in the manuscript, in the previous study, we performed detailed grading (Score 1-4) and distribution of dysplastic changes in the cerebellum. (ÄŒapo I, Hinić N, Lalošević D, et al. Vitamin C Depletion in Prenatal Guinea Pigs as a Model of Lissencephaly Type II. Veterinary Pathology. 2014;52(6):1263-1271.)
  • Additional image analysis like area/volume of lesion, myelination, neuron density and other we plan to analyse in future and present in more morphometric studies.
  • Analysis of the distribution and presence of intracranial haemorrhage was performed by an experienced neuropathologist (prof Dr Ivan ÄŒapo), and during analysis, histology slides were blinded!
  • In Table 3, we presented the distribution of intracranial haemorrhage, and we additionally performed statistical analysis, which you can find below.
  • Table 3. The distribution of intracranial haemorrhage

Group                   No.of fetuses

intra-axial haemorrhage (intraparenchymal)

extra-axial hemorrhage (subarachnoid)

cerebral cortex

hippocampus

thalamus and hypothalamus

cerebellum

Control

E1

E2

20

21

19

0

71

192,3

0

64

195,6

0

77

198,9

0

210

1911,12

0

0

1113,14

1 Con vs E1 p < 0,01; 2 Con vs E2 p < 0,001; 3 E1 vs E2 p < 0,001; 4 Con vs E1 p < 0,01; 5 Con vs E2 p < 0,001; 6 E1 vs E2 p < 0,001; 7 Con vs E1 p < 0,01; 8 Con vs E2 p < 0,001; 9 E1 vs E2 p < 0,001; 10 Con vs E1 p < 0,01; 11 Con vs E2 p < 0,001; 12 E1 vs E2 p < 0,001; 13Con vs E2 p < 0,001; 14 E1 vs E2 p < 0,001;

Statistically significant differences in the occurrence of hemorrhages among the groups were observed in the cerebral cortex (χ²(2) = 40.995, p < 0.001; Fisher’s exact test, Monte Carlo method), hippocampus (χ²(2) = 42.367, p < 0.001), thalamus and hypothalamus (χ²(2) = 40.995, p < 0.001), cerebellum (χ²(2) = 52.046, p < 0.001), and in the case of extra-axial (subarachnoid) hemorrhage (χ²(2) = 29.066, p < 0.001).

Comments 6: Hydroxyproline assay: Given the low collagen content of brain, discuss assay sensitivity, LOD/LOQ, and normalization. I wonder that Non-significant differences may reflect insufficient sensitivity rather than biology.

Response 6:

For hydroxyproline analysis, we use a protocol established by Hofman et al. (Hofman K, Hall B, Cleaver H, Marshall S. High-throughput quantification of hydroxyproline for determination of collagen. Anal Biochem. 2011 Oct 15;417(2):289-91.) We accept all your suggestions that non-significant differences may reflect insufficient sensitivity of the colourimetric assay rather than biology.

According to that, in the new version of the discussion, we added:

…analysed tissues. Although colourimetric analysis has been shown to be as sensitive as high-performance liquid chromatography (HPLC), more sensitive methods such as liquid chromatography-mass spectrometry (LC-MS) may be a better choice for brain tissue. However, the specificity…

Comments 7: Molecular endpoints: The Figure 6 legend shows “COL2” in places where the text focuses on Col4a1—please correct any nomenclature inconsistencies and ensure primer targets match reported genes.

Response 7: We corrected all nomenclature in the text.

Comments 8: Statistics and reporting: State biological n at the dam and fetus levels (accounting for litter as a random effect), not only the number of fetuses/sections. If multiple fetuses per litter were included, use models that adjust for clustering. Unify the analysis plan: specify normality/variance checks, ANOVA vs. non-parametric criteria, post-hoc corrections, and report effect sizes with 95% CIs alongside P values. Clarify when Mann–Whitney vs ANOVA was chosen, and apply a consistent multiple-comparison strategy. Figures should show individual data points, define what bars/points represent, and indicate blinding for image analyses.

Response 8:

We have entirely accepted your suggestion and improved it.

  • In the new version of the text, we added:

2.8. Statistical Analysis

Statistical analysis was performed using IBM SPSS statistical software, version 26.0 (IBM Corp., Armonk, NY, USA). Data were reported as the mean ± standard deviation (SD) or standard error of the mean (SEM). The normality of continuous variable distributions was assessed using the Kolmogorov-Smirnov and Shapiro-Wilk tests. In addition, the choice of alternative methods (nonparametric techniques) was based on variable type, coefficient of variance, and the results of the Levene test for homogeneity of variances. Because the assumptions for the chi-square test were not fully met, Fisher’s exact test (Monte Carlo method, two-sided) was applied to obtain precise significance values for differences in the frequency of intracranial haemorrhages among the experimental groups. A one-way analysis of variance (ANOVA) was used to compare groups for hydroxyproline concentrations. Post hoc testing for ANOVA was performed using Tukey’s test. For the analysis of mRNA relative concentration, an independent-samples T-test or its nonparametric alternative, the Mann–Whitney U test, was used. For each test, the corresponding effect size was calculated (Cohen’s d for t-tests, r for non-parametric tests). Where applicable, 95% confidence intervals (CIs) for the effect sizes were also reported to indicate the precision of the estimates. The difference between groups was considered statistically significant for a p-value less than 0.05 (p < 0.05).

Response 8:

We accept completely your mention about the distribution of n of dams and the distribution of fetuses according to the chosen method of analysis. According to that, we add a detailed explanation in the section' Materials and Methods' and an additional Table 4, which will be in the supplement.

We would like to note that we do not use multiple fetuses per litter for the biochemical and molecular analysis!

  • In the new version of the material and methods, we added:

Each group included 10 litters, each with 3 or 4 fetuses. From each group, 6 litters were randomly selected: 1 fetus per litter was taken for biochemical and molecular analysis, and 1 or 2 fetuses per litter for histological analysis. All fetuses from the remaining four litters in each group underwent histological analysis. Table 1 (Supplemental material) details the distribution of fetuses.

Table 4. Details about the distribution of fetuses.

Control group

E1 Vitamin C-deprived group

E2 Vitamin C-deprived group

Dam number

1

2

3

4

5

6

7

8

9

10

1

2

3

4

5

6

7

8

9

10

1

2

3

4

5

6

7

8

9

10

Number of fetuses

3

4

3

3

3

3

4

3

3

3

3

3

4

3

4

4

3

3

3

3

3

3

3

3

4

3

3

3

3

3

Biochemical analysis

1

1

1

1

1

1

-

-

-

-

1

1

1

1

1

1

-

-

-

-

1

1

1

1

1

1

-

-

-

-

Molecular analysis

1

1

1

1

1

1

-

-

-

-

1

1

1

1

1

1

-

-

-

-

1

1

1

1

1

1

-

-

-

-

Histology analysis

1

2

1

1

1

1

4

3

3

3

1

2

1

1

2

2

1

3

1

1

1

1

1

1

2

1

3

3

3

3

Interpretation and discussion

Comments 9 : The proposed link from vitamin C deprivation to PBM rupture leading to dysplasia and hemorrhage is plausible and consistent with prior models. However, causal inference would be stronger with quantification of PBM components (e.g., COL4A1, laminin) and objective hemorrhage metrics. Where such data are unavailable, please temper causal language and acknowledge perfusion–fixation as a possible contributor.

Response 9:

We accept all suggestions and add an explanation in the discussion.

In the pathogenesis of fresh haemorrhage, the leading cause is vascular wall disruption. High-quality tissue fixation is achieved by perfusion, during which fixative is pumped through the circulatory system. This technical step is crucial for the appearance of fresh haemorrhage in fragile blood vessels in the experimental groups. The absence of haemorrhage in the control group indicates that these vessels were not in the same condition after all. To improve the study design, we can use immersion fixation, which reduces pressure on the blood vessels.

Comments 10: Please also distinguishes clearly what replicates or extends prior work versus what is novel (e.g., the strict timing of complete deprivation and the staging of cerebellar dysplasia).

Response 10:

In the introduction's aim section, we add a sentence that clearly explains how our work extends the previous one.

… dysplasia [22]. Based on a previous study [22], we performed an extensive immuno-histochemical analysis of cerebrum and cerebellum, focusing on neural and glial differentiation in intensely dysplastic areas. We also conducted biochemical and molecular analyses in the animal model presented here.

Comments 11: I recommend that conclusions regarding hydroxyproline be appropriately qualified, given the assay’s constraints in brain tissue.

Response 11:

In question 6 (above), we discussed the assay's sensitivity limit for analysing brain tissue and what can be improved.

Comments 12: Comments on the Quality of English Language

Overall readability is adequate, but the manuscript needs light-to-moderate language editing. Recurring issues include: inconsistent terminology (e.g., PBM vs. BM; ROS used without definition), gene/protein nomenclature drift (COL4A1 vs. Col4a1; SLC23A1 vs. Slc23a1), article and preposition errors (“the” overuse; “effect on/of” mix-ups), tense inconsistency between Methods/Results, and punctuation/spelling slips (e.g., H2DCFDA vs. H2DCFDA/DCF; “perfusion–fixation” hyphenation; figure/axis labels with capitalization mismatches). Figure legends occasionally omit statistical details or units. A professional copy-edit to standardize style, fix typos, and harmonize nomenclature would be beneficial.

Response 12:

We enhanced the quality of the English language, standardised the style, corrected typos, and harmonised nomenclature.

Reviewer 3 Report

Comments and Suggestions for Authors

Vitamin C is one of the most important molecules for human health. It is necessary for correct collagen synthesis i.e., conversion of prolinÄ™ to hydroxyproline in the presence of Iron (Fe) ions and hydrogen peroxide. Moreover, Vitamin C is a common vitamin in the human diet and belongs to the water-soluble group, contrary to Vit ADE and K. The correct level of Vit C causes the correct collagen synthesis in different types. Authors should write some highlights of the above (collagen type and their location). Therefore, its lack in the diet leads to different disorders. The authors correctly used their model for Vit C deprivation investigation. The methodology from histology to statistical analysis through molecular techniques has been correctly described. The graphical frame is suitable, as well as the references, which are correctly selected. Even though the article is interesting, I have some critical remarks that must be answered:

- the medical mining and significance of Vit C reduction

- the results of Vitamin C overload for embryons

- the metabolism of Vit C and its interaction with Vit E

- can we observe the Vit C deprivation in humans (cases)

- the level of vitamin C in pregnant guinea pigs

- the level of Vit C in nutrients

- the significance of the presented results for physicians should be given in the abstract and conclusion.

In conclusion, after answering my question, the article can be considered.

Author Response

Dear reviewer,

We thank you and the reviewer for commenting on our manuscript, identifying its weakness, and, moreover, providing us the opportunity to strengthen our research and raise its quality.

Reviewer 3

Comments and Suggestions for Authors

Vitamin C is one of the most important molecules for human health. It is necessary for correct collagen synthesis i.e., conversion of prolinÄ™ to hydroxyproline in the presence of Iron (Fe) ions and hydrogen peroxide. Moreover, Vitamin C is a common vitamin in the human diet and belongs to the water-soluble group, contrary to Vit ADE and K. The correct level of Vit C causes the correct collagen synthesis in different types. Authors should write some highlights of the above (collagen type and their location). Therefore, its lack in the diet leads to different disorders. The authors correctly used their model for Vit C deprivation investigation. The methodology from histology to statistical analysis through molecular techniques has been correctly described. The graphical frame is suitable, as well as the references, which are correctly selected. Even though the article is interesting, I have some critical remarks that must be answered:

Comment 1: The medical mining and significance of Vit C reduction

Response 1:

We have entirely accepted your suggestion and improved it.

In the new version of Introduction, we added:

…synthesis [4]. This clearly defined pathomechanism represents a long-forgotten clinical entity called scurvy. Compromised collagen biosynthesis leads to systemic connective tissue breakdown, with classic symptoms such as bleeding gums, easy bruising, poor wound healing, and musculoskeletal pain. …

Comment 2: The results of Vitamin C overload for embryons

Comment 3: The metabolism of Vit C and its interaction with Vit E

The responses to comments 2 and 3 are presented in the text added to the discussion.

Response 2 and 3:

We have entirely accepted your suggestion and improved it.

In the new version of the discussion, we added:

In addition to its importance in collagen biochemistry, vitamin C has strong antioxidant properties that neutralise free radicals. At the same time, the regeneration of the oxidised form of vitamin E (a lipid antioxidant) to its active form also affects antioxidant activity. Applied alone (ref) or combined with vitamin E (ref), Vitamin C demonstrated a practical benefit of antioxidant potential through reducing oxidative stress-induced toxicity in embryos, as shown by improved blastocyst development.

Comment 4: Can we observe the Vitamin C deprivation in humans (cases)

Response 4:

We have entirely accepted your suggestion and improved it.

In the new version of the Introduction, we added:

Although forgotten, scurvy is again on the rise in developed countries, mainly due to poor diets high in processed foods and low in fresh fruits and vegetables. At-risk groups include alcoholics, smokers, individuals with eating disorders like anorexia, the elderly, and patients with chronic gastrointestinal diseases such as Crohn's disease. Smoking notably reduces vitamin C absorption and increases the body’s needs.

 Comment 5: The level of vitamin C in pregnant guinea pigs

Response 5:

Analysis of vitamin C concentration in serum or tissue was not performed. This limitation is noted in the discussion, which highlights the relevance of this analysis for enhancing the study.

In the new version of the discussion, we added:

… an increase in SVCT2 expression was also not observed [55].

Despite the compelling results, the study had limitations and can undoubtedly be improved by: volumetric and stereological brain analysis; Vitamin C serum and tissue level determination in both mothers and fetuses; quantification of specific types of collagen in tissues using the western blot or other methods; and extending the investigation beyond prenatal deprivation to examine its consequences during the postnatal period of life.

 Comment 6: The level of Vitamin C in nutrients

Response 6:

We have entirely accepted your suggestion and improved it.

In the new version of the Introduction, we added:

… musculoskeletal pain. Although forgotten, scurvy is again on the rise in developed countries, mainly due to poor diets high in processed foods and low in fresh vegetables and fruits like red peppers (125-150 mg in medium size), Broccoli (39,2 mg in 1/2 cup) or oranges (63.5 mg in medium size), strawberries (97 mg in 1 cup) and kiwifruit (70.5 mg in medium size). 

Comment 7: The significance of the presented results for physicians should be given in the abstract and conclusion.

Response 7:

We have entirely accepted your suggestion and improved it.

In the new version of the Abstract and Conclusion, we added:

Abstract ...  Conclusions: The presented morphological, biochemical, and molecular aspects of prenatal Vitamin C deprivation demonstrate strong translational potential for the human population. This opens a new perspective on the importance of this vitamin in fields such as public health, gynaecology, and obstetrics.

Conclusions ...  The results presented on the morphological, biochemical, and molecular aspects of Vitamin C deprivation confirm that guinea pigs are a unique experimental model for studying vitamin C deficiency. The observed neurodevelopmental changes and the potential for high translation of the results into the human population open a new perspective on the importance of this vitamin in areas such as public health, gynaecology, and obstetrics.

Round 2

Reviewer 1 Report

Comments and Suggestions for Authors

The manuscript was improved as I have suggested. 

Author Response

Dear reviewer,

Thank you once again for your support in improving our research and raising its quality.

Best regards,

Prof. dr Ilija Andrijevic

Reviewer 2 Report

Comments and Suggestions for Authors

General Comments
The manuscript has been revised carefully and most prior concerns have been addressed with clearer methods, improved transparency, and more consistent wording. The overall presentation is substantially better.

Special Comments

  1. Please continue to frame the main findings as associations rather than causality. The proposed chain “vitamin C deprivation → PBM disruption → dysplasia/hemorrhage” is plausible but lacks quantitative corroboration. Although the limitations are acknowledged in the text, please ensure the tone remains consistently tempered across the Abstract, Results, Discussion, and Conclusions.

  1. Re-check the Abstract word count to ensure full compliance with the journal’s limit and adjust if necessary.

Author Response

Dear reviewer,

We thank you and the reviewer for commenting on our manuscript, identifying its weakness, and, moreover, providing us the opportunity to strengthen our research and raise its quality.

Reviewer 2

General Comments
The manuscript has been revised carefully and most prior concerns have been addressed with clearer methods, improved transparency, and more consistent wording. The overall presentation is substantially better.

Special Comments

Comment 1: Please continue to frame the main findings as associations rather than causality. The proposed chain “vitamin C deprivation PBM disruption dysplasia/hemorrhage is plausible but lacks quantitative corroboration. Although the limitations are acknowledged in the text, please ensure the tone remains consistently tempered across the Abstract, Results, Discussion, and Conclusions.

Respond 1: We accept your suggestion and presented the Vitamin C findings as associations rather than causality.

In the old version of the abstract, it was:

…deficiency. However, few studies, like ours, have directly linked vitamin C deprivation and the consequential weakening of the basement membrane with significant alterations in brain structure. Methods: …

In the new version of the abstract, we added:

 ...vitamin C deficiency. However, few studies, such as ours, indicate a possible link between vitamin C deprivation and the consequent weakening of the basement membrane, leading to significant alterations in brain structure. Methods:...

In the old version of the results, it was:

... 75]). Additionally, vitamin C deprivation affected the expression of the Slc23a1 gene. Both...

In the new version of the results, we added:

…75]). Additionally, in the vitamin C-deprived group, SLC23A1 expression was affected. Both…

In the old version of the discussion, it was:

…fetal growth [25, 26]. The results of our study indicate that earlier and prolonged Vitamin C deprivation in the E2 group leads to extensive changes in the fetuses, including weight loss [25] and the development of malformations. The observed

In the new version of the discussion, we added:

…fetal growth [25, 26]. The results of our study indicate that earlier and prolonged Vitamin C deprivation in the E2 group suggests a possible association with extensive fetal changes, including weight loss [25] and the development of malformations. The observed…

In the old version of the conclusions, it was:

… The results presented on the morphological, biochemical, and molecular aspects of Vitamin C deprivation confirm that guinea pigs are a unique experimental model for studying vitamin C deficiency. The …

In the new version of the conclusions, we added:

...The results presented on the morphological, biochemical, and molecular aspects of Vitamin C deprivation indicate that guinea pigs have a high potential as a unique experimental model for studying vitamin C deficiency. The...

Comment 2:

Re-check the Abstract word count to ensure full compliance with the journal’s limit and adjust if necessary.

Respond 2:

We rechecked the Abstract word count. Now it is fully compliant with the journal’s limit.

Reviewer 3 Report

Comments and Suggestions for Authors

The Authors corectly ansverd to my questions. I can recoment the current version for publication in the Nutrients journal.

Author Response

(The authors gave the same response as above.)
